# Effect of High Pressure Spark Plasma Sintering on the Densification of a Nb-Doped TiO2 Nanopowder

**Alexandre Verchère [1], Sandrine Cottrino [2,*], Gilbert Fantozzi [2], Shashank Mishra [1], Thomas Gaudisson [3], Nicholas Blanchard [3], Stéphane Pailhès [3], Stéphane Daniele [4] and Sylvie Le Floch [3,*]**

[1] IRCELYON, Université Lyon1-CNRS, UMR 5256, 2 Avenue Albert Einstein, 69626 Villeurbanne, France; alexandreverchere.av@gmail.com (A.V.); shashank.mishra@ircelyon.univ-lyon1.fr (S.M.)

[2] MATEIS, INSA-Université Lyon1-CNRS, UMR 5510, INSA de Lyon, 69621 Villeurbanne, France; sandrine.cottrino@insa-lyon.fr (S.C.); gilbert.fantozzi@insa-lyon.fr (G.F.)

[3] Institut Lumière Matière, Université Lyon1-CNRS, UMR 5306, Université de Lyon, 69622 Villeurbanne, France; thomas.gaudisson@univ-lyon1.fr (T.G.); nicholas.blanchard@univ-lyon1.fr (N.B.); stephane.pailhes@univ-lyon1.fr (S.P.); sylvie.le-floch@univ-lyon1.fr (S.L.F.)

[4] C2P2, Université Lyon 1—CPE Lyon-CNRS, UMR 5265, 43 Bvd du 11 Novembre 1918, 69616 Villeurbanne, France; stephane.daniele@univ-lyon1.fr

\* Correspondence: sandrine.cottrino@insa-lyon.fr (S.C.); sylvie.le-floch@univ-lyon1.fr (S.L.F.); Tel. +33-472-431-402(S.L.F.)

**Abstract:** Sintering under pressure by means of the spark plasma sintering (SPS) technique is a common route to reduce the sintering temperature and to achieve ceramics with a fine-grained microstructure. In this work, high-density bulk TiO2 was sintered by high pressure SPS. It is shown that by applying high pressure during the SPS process (76 to 400 MPa), densification and phase transition start at lower temperature and are accelerated. Thus, it is possible to dissociate the two densification steps (anatase then rutile) and the transition phase during the sintering cycle. Regardless of the applied pressure, grain growth occurs during the final stage of the sintering process. However, twinning of the grains induced by the phase transition is enhanced under high pressure resulting in a reduction in the crystallite size.

**Keywords:** nanopowder TiO2; high-pressure spark plasma sintering; grain growth; densification rate

## 1. Introduction

Nb-doped Titanium dioxide (Nb-TiO2) is a promising cheap, chemically stable and nontoxic transition metal oxide for high temperature thermoelectric (TE) devices working in oxidizing environments. A good TE efficiency requires materials with low thermal conductivity and high electrical conductivity. While a small amount of Nb (2–4 at.%) increases the electrical conductivity of TiO2 by several orders of magnitude, it has been recently shown that the thermal conductivity of bulk nano-structured Nb-TiO2 obtained by spark plasma sintering (SPS) is significantly lowered by decreasing the average grain size [1]. A value of 2.5 W/mK has been observed at 900 K for an average grain size of 170 nm, while the value in the single crystal is of about 4 W/mK. SPS has proven its efficiency in fast densification with limiting grain growth [2,3]. However, the role of the applied pressure remains unclear [4–6]. This work provides a detailed investigation of the sintering mechanism of Nb-TiO2 during the SPS process and of the effect of the high pressure with the aim of identifying the parameters that control the average grain size in the nano-structured materials.

Our aim is to investigate whether high-pressure SPS can produce dense ceramics with finer nanostructures.

## 2. Materials and Methods

### 2.1. Synthesis of Niobium-Doped TiO$_2$ Nanoparticles.

All syntheses were carried out under argon using standard air-free techniques. The reagents, [Ti(OEt)$_4$]$_3$ and [Nb(OEt)$_5$]$_2$(Strem Chemicals, 99.9%) were distilled prior to use. Anhydrous pentane was obtained using a solvent drier, MB SPS-800 (solvent purification system), from MBRAUN and stored under argon over 4 Å molecular sieves. In a typical sol–gel synthesis procedure for the preparation of a TiO$_2$ nanopowder doped with 2 mol% Nb: 17.00 g (24.84 mmol) of [Ti(OEt)$_4$]$_3$ taken in 15 mL of pentane were mixed with 0.47 g (0.75 mmol) of [Nb(OEt)$_5$]$_2$ under stirring at room temperature. This solution was added dropwise to 150 mL of water with 2.40 g (7.45 mmol, 0.1 eq/Ti) of (N$^t$Bu$_4$)Br at 100 °C, and the medium was stirred for 1 h at 150 °C. The suspension was then centrifuged to yield a white solid. The as-prepared precipitate was washed by 3 × 50 mL of deionized water and 50 mL of ethanol, and dried at 80 °C for 12 h. Then, to remove all the organic compounds, the powder was calcinated at 500 °C for 4 h under air. After the calcination, the powder crystallized under the anatase phase (JCPDS no. 00-021-1272) with a brookite fraction of 5%. The average crystallite size, determined by Rietveld refinement, was around 10 nm. This crystallite size corresponds to the particle size observed by transmission electronic microscopy (TEM) (Figure 1). These crystallites are agglomerated in aggregates smaller than 1μm, as measured by laser granulometry.

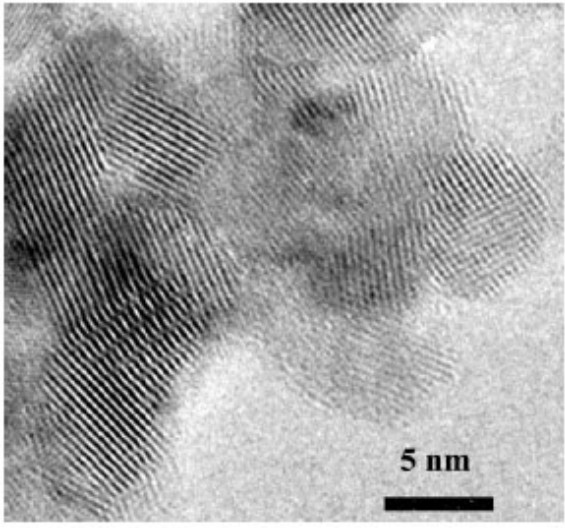

**Figure 1.** TEM image of TiO$_2$: 2 mol% Nb nanopowder calcined at 500 °C for 4 h.

### 2.2. SPS Densification.

The titanium dioxide powders, after the calcination, were sintered by using spark plasma equipment (SPS) HPD 25 (FCT Systeme GmbH, Germany). The powdered samples were loaded in a tungsten cemented carbide (WC-Co) die (10 mm inner diameter) lined with a graphite foil (0.35 mm thick). The sintering temperature was regulated by a K-type thermocouple (1 mm diameter) introduced in the WC die. The pulse patterned consisted of two pulses lasting 10 ms followed by one pause period of 5 ms.

Four experiments were performed in order to observe the effect of pressure on the sintering behavior of the anatase powder. The powder was first pressurized up to the targeted pressure (76, 200, 300 and 400 MPa). Then, the same heating cycle was applied for each experiment (heating rate n°1-heating rate n°2-sintering temperature-dwell time): 100 °C/min up to 680 °C–20 °C/min up to 700 °C-10 min. After the dwell time, the heating power was shut off and the pressure was released.

Subsequently, for a better understanding of phenomena, two samples were sintered under 400 MPa following the same heating rates up to a lower sintering temperature without dwell time: 450 °C and 600 °C.

Two samples sintered at 76 MPa by heating 10 min at 850 and 960 °C with heating rates 10 and 100 °C/min, respectively, are also referred for comparison.

*2.2 Characterization Techniques.*

The density of the sintered samples was measured using Archimedes method with distilled water as a liquid medium. The values for the relative densities were calculated assuming a theoretical density of titanium dioxide doped with 2 mol% of niobium. The calculated theoretical density is 3.925 g·cm$^{-3}$ for the anatase phase and 4.267 g·cm$^{-3}$ for the rutile phase. X-ray diffraction (XRD) was used to analyse the crystallographic phase of the powder and the sintered sample. A Bruker D8 Advance A25 diffractometer (Cu k$_\alpha$ radiation at 0.154184 nm) equipped with a Ni filter and 1-D fast multistrip detector (LynxEye, 192 channels on 2.95°) was used. The diffractograms were collected at 2Θ with steps of 0.02° from 15 to 85° (2Θ) for a total acquisition time of 120 min. The data analyses were carried out by Rietveld refinement (Fullprof Suite package) [7] using Thompson Cox Hastings (modified pseudo-Voigt) and instrumental resolution functions.

The observation of microstructures and the determination of grain size were performed using scanning electron microscopy (SEM) and image analysis software (ImageJ). The average size was obtained by measuring about 100 grains. Transmission electronic microscopy (TEM) of prepared thinned blades was performed using a MET JEOL2100.

# 3. Results and Discussion

*3.1 Effect of Pressure on Densification and Phase Transition*

To study the influence of pressure on the densification and the phase transition phenomena, four TiO$_2$:2%Nb pellets were sintered by SPS at 76, 200, 300 and 400 MPa at a constant temperature of 700 °C according to the thermal cycle described in the experimental part. Sintering of the pre-calcined powders resulted in pellets (density > 85%) of 10 mm diameter and 2 mm thick, with a metallic blue-black color. The characteristics of the pellets are presented in Table 1. It can be observed from XRD profiles (Figure 2) that the phase transition shifted to lower temperatures as the pressure increased. Indeed, at 76 MPa—700 °C, the pellets were mainly composed of anatase phase, while at 200 MPa—700 °C, they mainly consisted of rutile phase (Figure 2). For pure TiO$_2$ nanoparticles, the anatase to rutile transformation occurred within 600–850 °C [8]. The transformation from anatase to rutile involved a volume decrease of 8.7%, which was promoted by the application of pressure [9]. On the other hand, according to our previous work [10], Nb doping (2% mol.) delays the transition temperature from 700 to 800 °C during the SPS sintering at 76 MPa.

**Table 1.** Characteristics of the different pellets produced in this study. The heating rate is 100 °C/min for all samples, except for the sample "850 °C–76 MPa" which was heated at 10 °C/min. The mean grain size and the corresponding standard deviation, D90 (90% of the particles are below this size), D50 (median) are evaluated from the SEM images. The crystallite sizes are extracted from the XRD patterns fitted using Fullprof Suite package [7].

| Pellets | Phases (A + R = 100%) | Relative Density (%) | Mean Grain Size (nm) | Standard Deviation (nm) | Crystallite Size (nm) | | D90 (nm) | D50 (nm) |
|---|---|---|---|---|---|---|---|---|
| | | | | | Anatase | Rutile | | |
| 700 °C–76 MPa | **69.6% A** | 84.8 | 39 | 12 | 31 | 66 | 57 ± 2 | 38 ± 1 |
| 700 °C–200 MPa | **98% R** | 96.2 | 260 | 53 | | 86 | 319 ± 7 | 258 ± 2 |
| 700 °C–300 MPa | **100% R** | 95.3 | 287 | 101 | | 93 | 455 ± 7 | 306 ± 3 |
| 700 °C–400 MPa | **100% R** | 96.0 | 322 | 98 | | 62 | 428 ± 7 | 272 ± 2 |
| 600 °C–400 MPa | **98.9% R** | 94.2 | 168 | 73 | | 92 | 277 ± 6 | 165 ± 6 |
| 450 °C–400 MPa | **95.4% A** | 89.7 | 22 | 5 | 17 | | 28 ± 2 | 22 ± 1 |
| 850 °C–76 MPa | **100% R** | 89.0 | 159 | 70 | | 136 | 225 ± 3 | 152 ± 2 |

| 960 °C–76 MPa | **100% R** | 96.3 | 252 | 158 | 92 | 448 ± 4 | 200 ± 4 |
|---|---|---|---|---|---|---|---|

**Figure 2.** XRD patterns of the materials obtained after spark plasma sintering (SPS) cycle up to 700 °C at the different pressures. The peaks are identified as the anatase phase (**a**) and the rutile phase (**r**) in the samples. (**g**) Corresponding to graphite residue (originating from the graphite foil placed between the WC mold and the sample during sintering) present on the surface of a sample.

The SPS dilatograms for these four samples are shown in Figure 3a–d. The displacement rate of the piston (related to the sample shrinkage rate) shows that for the low pressure (76 MPa), the densification was slow and extended over a wide temperature range (from 350 to 700 °C). A small part of the sample was transformed into rutile. For an intermediate pressure (200 MPa), the densification started near 280 °C and extended up to 700 °C. A large part of the sample was transformed into rutile. As the pressure increased, the first stage of densification took place at a lower temperature and the transformation of the anatase phase to the rutile phase was favored, but no phase transition signature was observed in the dilatogram during the temperature rise.

For the higher pressures (300 and 400 MPa), the dilatograms exhibited two stages of densification around 380 and 650 °C. For the sample sintered at 300 MPa, the maximum displacement rate of the first densification stage was observed at 385 °C. At 400 MPa, three phenomena could be distinguished. The first and the second densification stages show the maximum displacement rates at 365 and 620 °C, respectively. An interesting phenomenon can be noted for the sintering at 400 MPa. At 546 °C, we observed a sharp jump with a piston displacement of 0.15 mm which may correspond to the transition from the anatase phase (theoretical density = 3.925 g·cm$^{-3}$) to the rutile phase (theoretical density = 4.267 g·cm$^{-3}$)—i.e., $\Delta V/V = -8$ %. Indeed, we placed about 1 g of the powder in the matrix; a phase change caused a variation of the volume of about 0.02 cm$^3$. This implies a piston displacement of 0.26 mm. This variation is purely indicative because we are uncertain about the exact mass of the powder, the density of the pellets, and we did not take into account the thermal expansion of the lattice. The phase transition rate at 400 MPa was very fast and certainly occurred at a lower temperature as compared to sintering at 300 MPa where this peak was not observed. At 200 and 300 MPa, the transition phase occurred but was more gradual. In fact, the anatase–rutile transition was

reconstructive, and the total transformation took place slowly between 600 and 700 °C under moderate pressure [8].

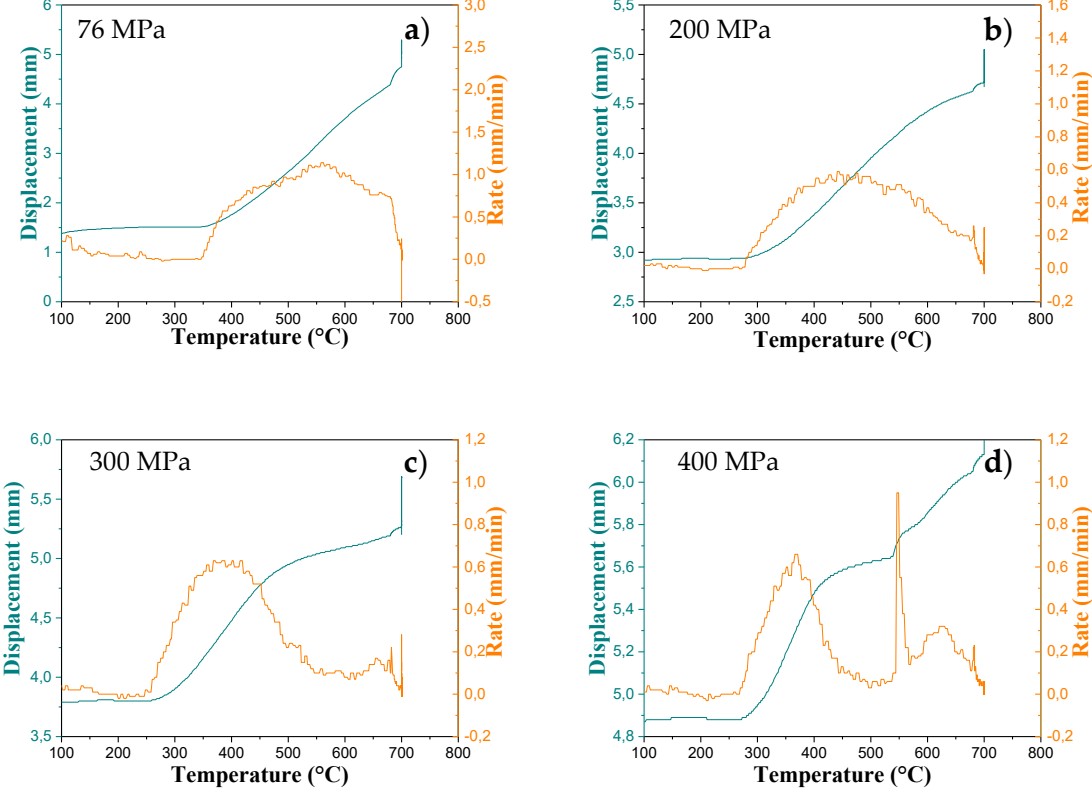

**Figure 3.** Sintering cycles of TiO$_2$:2%Nb nanoparticles (pre-calcined at 500 °C/4 h) up to 700 °C under different pressures: (**a**) 76, (**b**) 200, (**c**) 300 and (**d**) 400 MPa.

In order to understand the significant piston displacement rate observed in Figure 3d, two sintering experiments were performed at 400 MPa. The first sintering was interrupted at 450 °C, before the sharp peak and after the first densification peak. The second sintering was stopped at 600 °C immediately after the sharp peak (Figure 3d). The XRD patterns (Figure 4) obtained at the pressure of 400 MPa could be identified mainly with the anatase phase for 450 °C and the rutile phase for 600 °C. This result (Figure 4) clearly shows that the sharp jump observed around 546 °C was related to the sudden transformation of the anatase to the rutile phase. In consequence, the two densification stages observed during the sintering at 400 MPa can be attributed to the sintering of the anatase phase around 365 °C and then of the rutile phase around 620 °C. After the first stage, at 450 °C, the relative density of the pellet that mainly comprised the anatase phase was equal to 89.7%. After the second densification stage, at 700 °C, the relative density of the pellet reached 96%, and the anatase phase was totally transformed to the rutile phase.

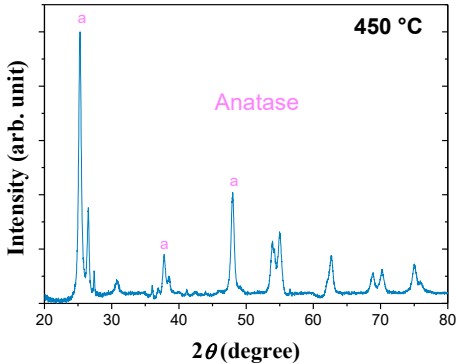
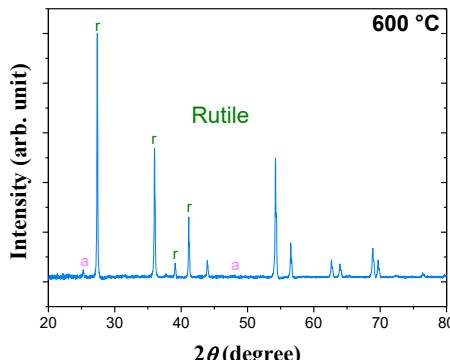

**Figure 4.** XRD patterns of the materials obtained after SPS cycles up to 450 and 600 °C under 400 MPa.

The Figure 5 presents the evolution of the temperature corresponding to the maximum of the piston displacement speed, observed on the dilatograms, as a function of the applied pressure. A decrease of about 200 °C was induced when the pressure increased from 76 to 400 MPa. The increase in the pressure accelerated the rates of the densification and the phase transformation (Figure 3) and lowered their occurrence temperature (Figure 5), which led to dissociate the different densification steps (anatase then rutile) and to distinguish the phase transition.

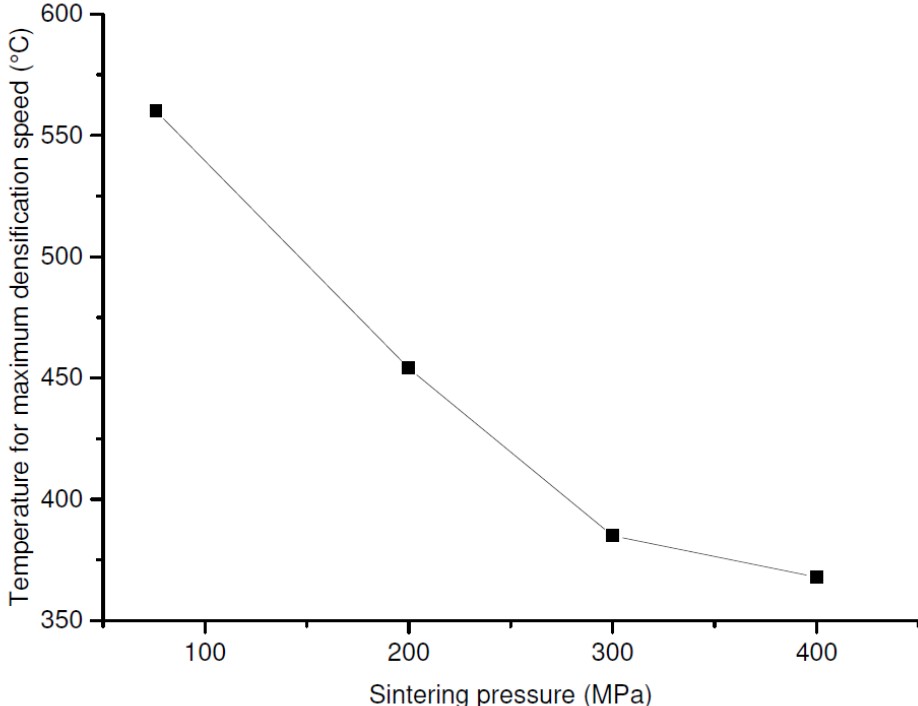

**Figure 5.** Effect of the pressure on the temperature showing the maximum densification rate.

At a constant sintering temperature (here 700 °C), as the sintering pressure increases, the anatase phase progressively transforms to the rutile phase (Figure 2). After sintering at 76 MPa, two phases coexisted, and a density of 3.42 g·cm$^{-3}$ was measured. After sintering at 200 MPa, a small anatase peak could still be seen, but the pellet was mostly rutile with a density of 4.10 g·cm$^{-3}$. Above this pressure, the densities no longer change and the diffractograms showed a single rutile phase. This result confirms that with increasing pressure, the phase transition is shifted to lower temperatures, as already reported by Liao et al. [6]. They observed by hot pressing that the transformation is lowered to 400 °C under a pressure equal to 1.5 GPa. It can be noticed that Maglia et al. [5] obtained different

results by high pressure field assisted rapid sintering of the anatase phase using a different sintering cycle than ours.

### 3.2. Effect of Pressure on Grain Size

In order to measure the grain size, SEM observation was performed on the pellets sintered at 700 °C with a heating rate of 100 °C/min and several pressures of 76, 200, 300 and 400 MPa. The images are shown in Figure 6. The pellet sintered at 76 MPa had a very fine microstructure with a mean grain size of 39 nm. Coarsening of grains was obvious in sintered samples under the higher pressures, leading to a mean grain size higher than 260 nm. Figure 7 shows the grain size distributions (GSD) extracted from the analysis of the SEM images (Figure 6). As the pressure increased, the GSD broadened and deviated increasingly from a Gaussian shape. This suggests that abnormal grain growth (AGG) occurs when pressure is equal to or higher than 300 MPa. AGG is known to lead to GSD broadening, which can ultimately produce a bimodal distribution [11]. In contrast, for samples sintered at 76 and 200 MPa, the GSD shape suggests a normal grain growth (NGG). AGG may occur only at the final sintering stage when the relative density has reached at least 95% [12].

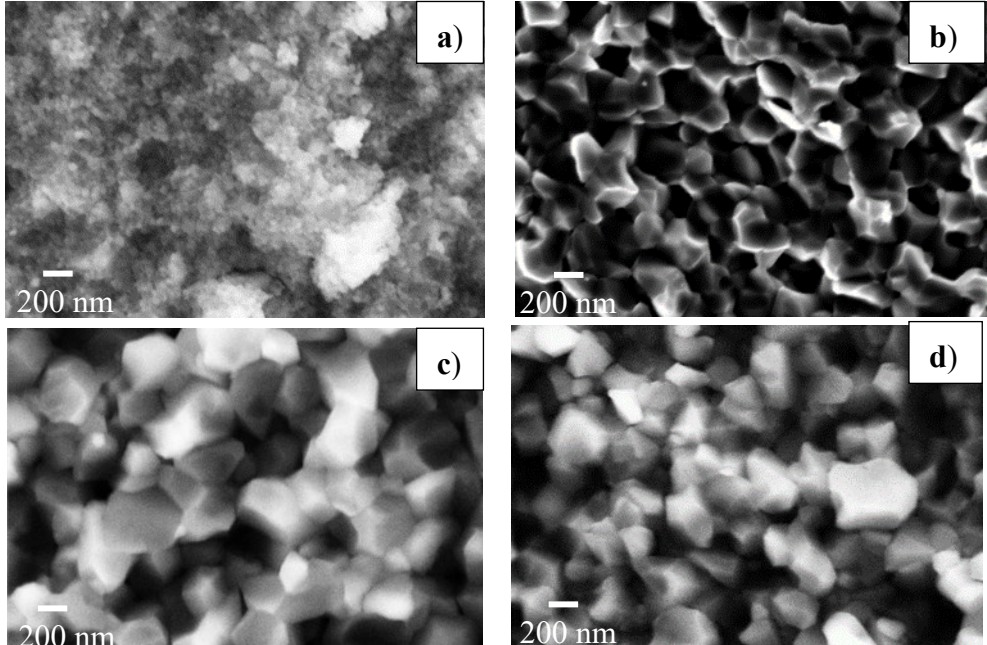

**Figure 6.** SEM images of the sintered pellets at 700 °C with the different pressures of (**a**) 76, (**b**) 200, (**c**) 300 and (**d**) 400 MPa.

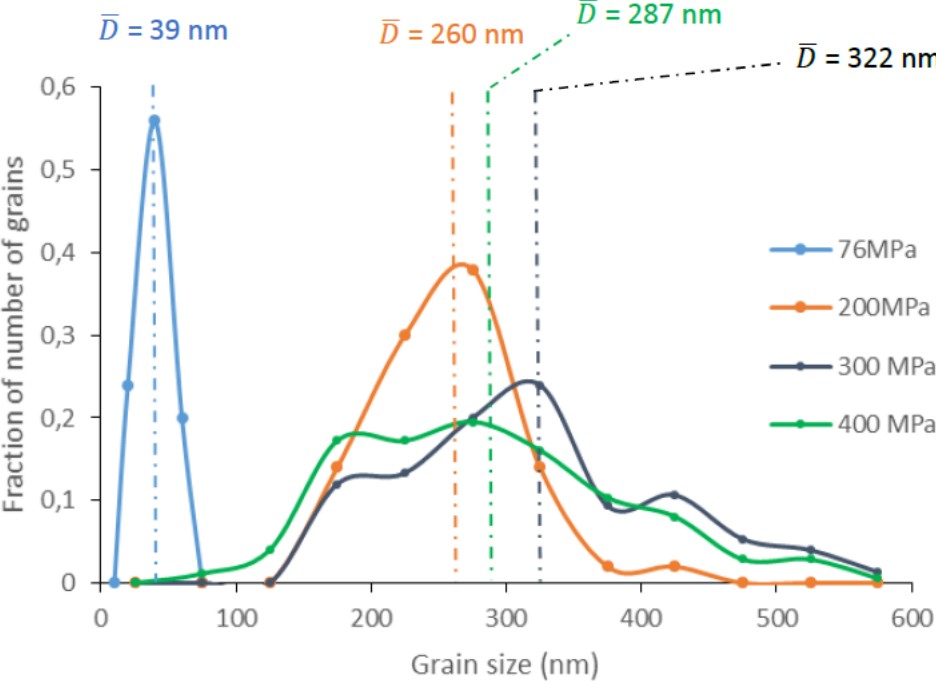

**Figure 7.** Grain size distributions of the samples sintered at 700 °C for 10 min under different pressures. Grain size is the grain diameter measured on the SEM images (50 to 175 values per sample). The data are discretized in twelve 50 nm width intervals. For the clarity of the graphic representation, each interval is represented by its center value. The average grain size, $\bar{D}$, and the standard deviation, SD, are reported for each sample.

The grain size and relative density of the pellets sintered at 700 °Care plotted in Figure 8 as a function of pressure. It shows that for the same density, the grain size increases with pressure. To compare with the density and the microstructure of a pellet sintered at a higher temperature (960 °C) but at the lower pressure (76 MPa), the density and grain size values are added in the graph. It can be noticed that the relative density (96%) and the mean grain size of this pellet were similar to those of the pellet synthesized at 700 °C under 200 MPa. Under the higher pressures (300 and 400 MPa), the sintering at 700 °C did not enhance the densification but led to AGG. These results confirm that AGG starts at the final sintering stage, when the relative density has reached about 95%.

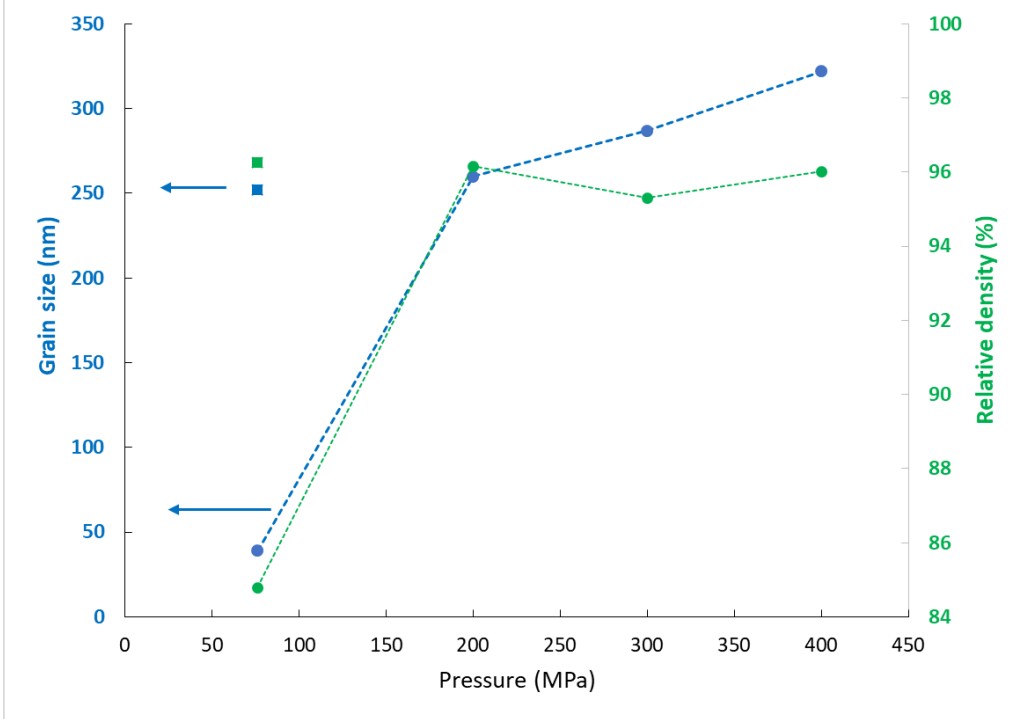

**Figure 8.** Evolution of grain size and relative density as a function of pressure for a thermal cycle up to 700 °C.

Figure 9 shows the evolution of grain size during the SPS at an equal pressure of 400 MPa, between 450 and 600 °C. With a temperature difference of 150 °C, the grain size increased from 22 to 168 nm. It seems that the phase transition led to an increase in grain size.

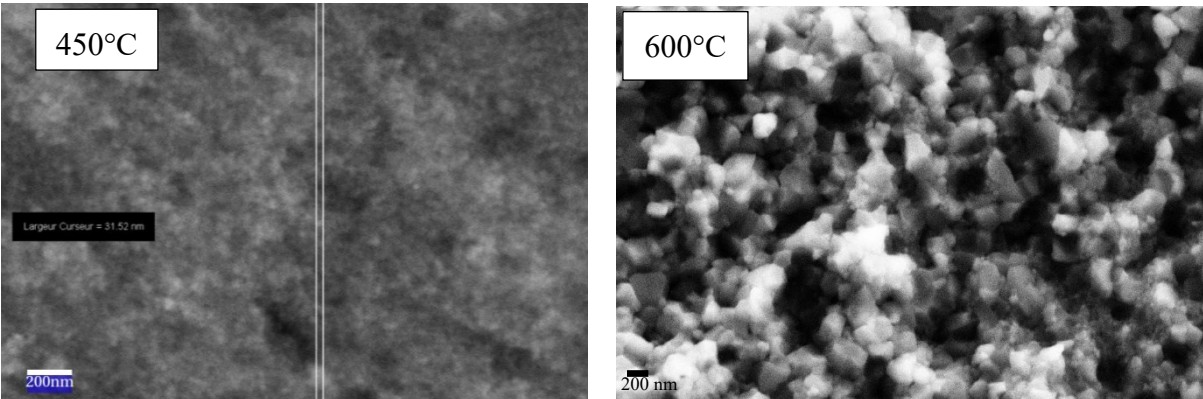

**Figure 9.** SEM images of niobium-doped TiO$_2$ after SPS at 400 MPa in a WC matrix. The influence of phase transition on grain size is clearly visible in the SEM images.

This effect of the phase transition can be observed in Figure 10, representing the evolution of grain size as a function of density for the isobaric sintering cycles: 76 and 400 MPa. We have shown (Figure 4) that at 400 MPa, the transition occurred around 500 °C. Three pellets were sintered under 76 MPa at three different sintering temperatures (700, 850 and 960 °C) (Table 1). XRD analysis was performed on each pellet, and it appears that transition phase occurred around 700 °C. At 850 °C, the pellet consisted only of the rutile phase. It appears that a significant grain growth occurred around the phase transition temperature range.

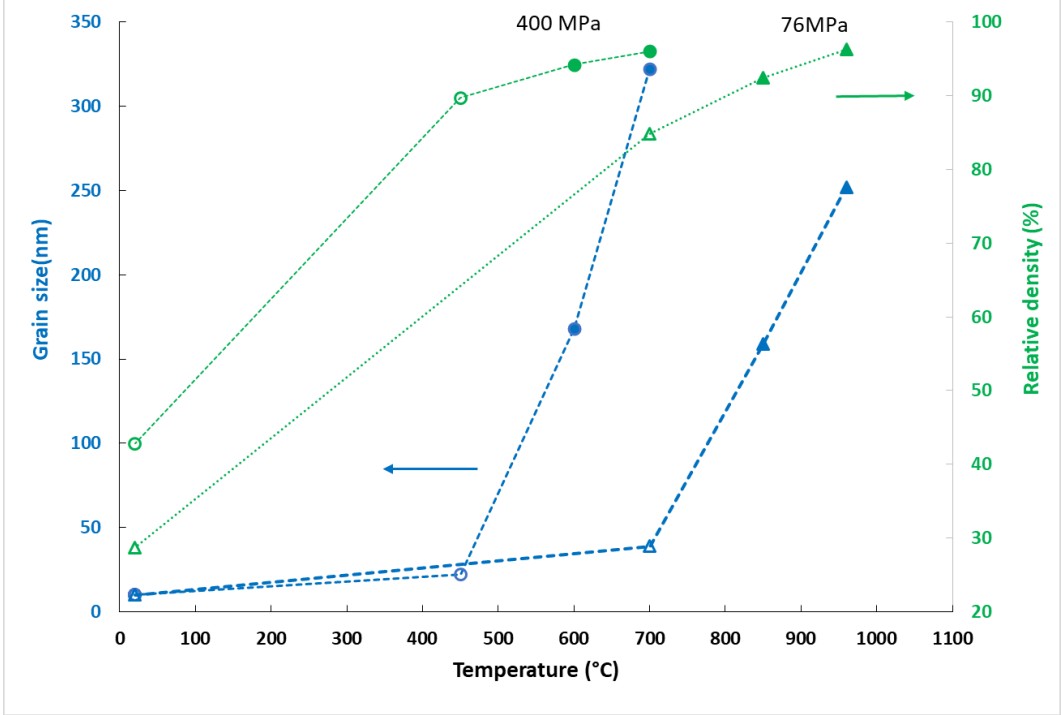

**Figure 10.** Evolution of the grain size and the density as a function of the temperature at 2 constant pressures: 76 (△) and 400 MPa (○). Open and closed symbols indicate if the sample is mainly crystallized in the anatase phase or the rutile phase, respectively.

High applied pressure causes strong constraints on the grains and thus a higher density of the initial compacts. As the temperature increases, thermal energy drives the phase transition. It is clear that highly constrained grains lead to a transition at lower temperature. As soon as the transition has occurred and the densification is partially achieved, grain growth accelerates regardless of the pressure. Pellets sintered in the rutile stability region, at 400 MPa and 600 °C and at 76 MPa and 850 °C had the same mean grain size of 160 nm and the same relative density (94%). However, at lower temperature, in the anatase stability region, a pressure as high as 400 MPa could densify the anatase up to 90% with very a limited grain growth (22 nm). These observations suggest that the grain growth is not driven by the same phenomena at 76 and 400 MPa. At 76 MPa, grain growth was governed by diffusion phenomena. At 400 MPa, it was managed by the grain boundary sliding and grain rotation generated by mechanical stress. For pellets with a density lower than 95%, sintering at 400 MPa allowed compacts with finer microstructures to be obtained. The gain in density is due to the very dense grain packing obtained by the high initial pressure. They were arranged as compactly as possible. Once the 95% relative density was overcome, it was observed that the compact sintered at 400 MPa had a coarser microstructure for the same density as compact sintered at 76 MPa due to AGG. Below 95% densification, grain coarsening appeared to be a direct consequence of phase transformation via nucleation and fast grain growth. The transition to the rutile phase was accompanied by significant grain growth resulting in large rutile grains and small anatase ones (Figure 12b). The hypothesis of rutile nucleation at the boundary between anatase particles can explain the formation of large rutile grains by the coalescence of two or more anatase grains [8]. In Figure 11, we report the grain size as a function of the relative density for the eight samples. The relative density of the powder compacted at 1000 MPa at room temperature is also reported. This was measured in compression using a Zwick testing machine. Relative density for the compact at 76 and 200, 300, 400 MPa, not reported on the graph, were equal to 29, 35, 39 and 43%, respectively.

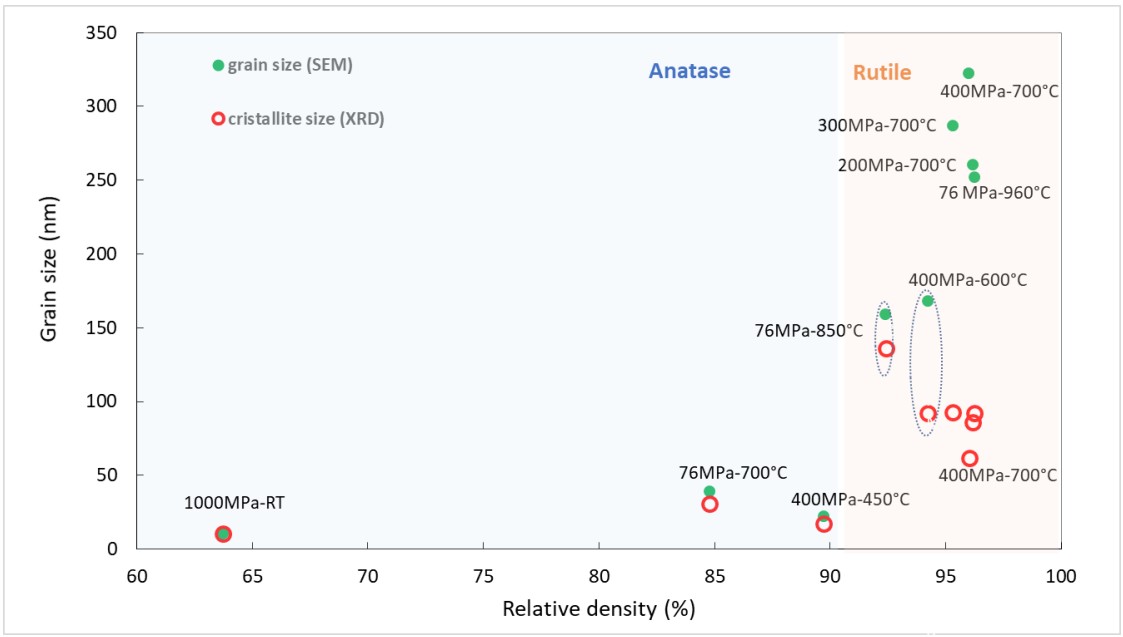

**Figure 11.** Grain size and crystallite size versus relative density for all samples.

It is clear from Figure 11 that the phase transition induced a jump in grain size from 30 to 170 nm regardless of the applied pressure. Then, grain growth accelerated when the density exceeded 95% independently of the process pressure. The higher grain growth rate observed for rutile grains is not inherent in the nature of the phase. In fact, the activation energy for grain growth is known to be five times smaller for anatase nanoparticles than for rutile nanoparticles [13]. The observed grain growth acceleration appears to be related to the relative density, as shown by Bernard-Granger's study on spark plasma and hot-pressure sintering of zirconia [14]. This could be related to the pore pinning effect [15]. In fact, as long as the sample contains dispersed pores, they pin the grain boundaries and prevent their migration, reducing the rate of the grain growth. When the porosity decreases to less than 5%, the pinning of the pores decreases, which dramatically accelerates the grain growth.

As mentioned above, high pressure sintering increased the densification rate but also the grain growth rate at the end of the sintering cycle, inducing an AGG. This acceleration is induced by grain boundary sliding and grain rotation. This stress-enhanced dynamic grain growth has been demonstrated in the work of Barak Ratzker et al. [16] on the sintering of transparent alumina. Sliding and rotation can lead to grain coalescence [17] for high temperature deformation of dense, fine-grained materials. The rotation of one grain relative to the other occurs by sliding, so that the crystallographic atomic planes of the two neighboring grains become identical. The grain boundary is erased, and a single grain is formed. This mechanism contributes to enhanced grain growth during sintering at low temperatures under high pressure. Large grains (400 nm) with angular grain-boundary curvatures were observed by TEM in the sample sintered at 700 °C under 400 MPa (Figure 12a). These unstable grains may be due to the slow diffusion rates observed at relatively low temperatures. As shown in Figure 10, at 400 MPa, AGG occurred in a temperature range of 500 to 700 °C, which are low temperatures to activate diffusion phenomena.

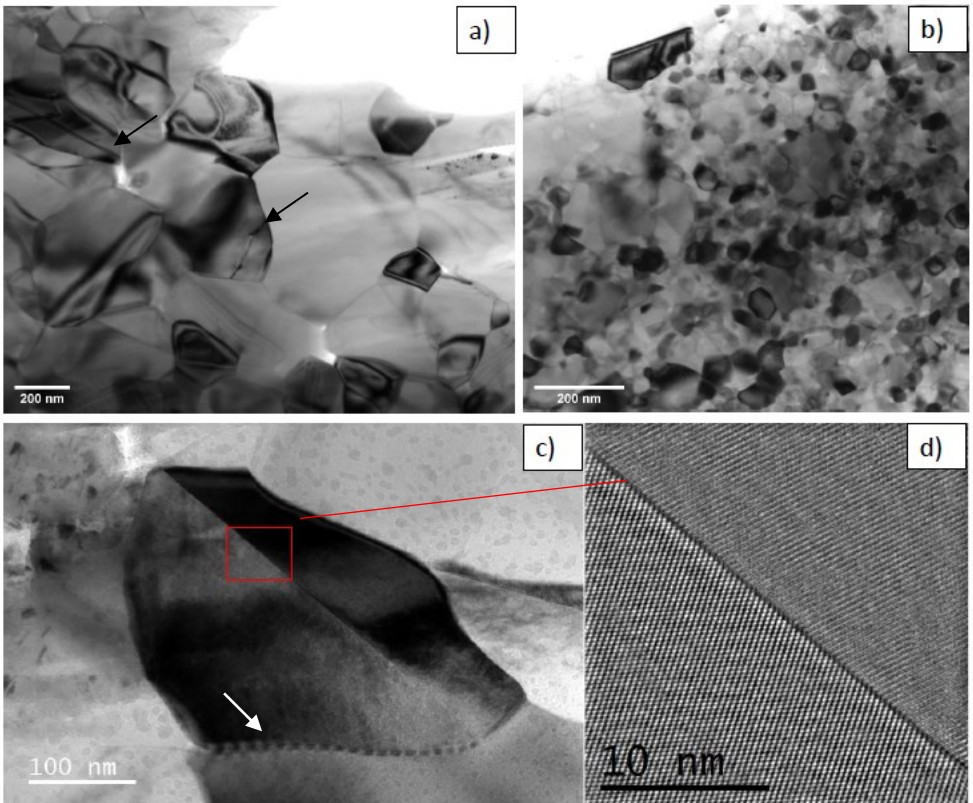

**Figure 12.** TEM images of samples sintered at 700 °C under 400 MPa. The black arrow shows fractured grains (**a**) and 76 MPa (**b**); grain comprised of twinned crystals, observed in the sample sintered under 400 MPa. The white arrow shows an intragranular dislocation fold originating from a triple point (**c**); high resolution TEM image of the twin boundary (**d**).

A more accurate observation showed that the majority of the grains consisted of twinned crystals (Figure 12c,d). Some of them were fractured along the twin boundaries (Figure 12a). The twinning should be due to anatase–rutile transformation, but it was clearly enhanced by increasing pressure. In fact, a high stress is known to promote twinning. In addition, the higher pressure (400 MPa) applied during sintering induced an abrupt phase transition with volume reduction of 8%, which generated a strong additional stress leading to twinning and fracture of the rutile grains. Intragranular dislocation folds from triple points are also visible in this sample (Figure 12c). These kind of defects have been reported on fine-grained $Y_2O_3$-tetragonal zirconia ceramic by Bernard-Granger et al. [18]. They explain this microstructure as the result of grain boundary sliding during high temperature creep deformation under 100 MPa.

In our case, the twinning and fracture of grains were consistent with the evolution of the crystallite size extracted from the X-ray diffractogram analysis (Table 1 and Figure 11). Above a relative density of 95%, the crystallite size was smaller than the grain size observed by SEM. This difference was maximum for the sample sintered at 400 MPa–700 °C, which underwent colossal stress due to the fast phase transition combined with the high applied pressure. The stress generated can induce dislocations and even rupture of large rutile grains. These results are consistent with those reported by Maglia et al. [5], who observed a decrease in the rutile crystallite size with increasing sintering temperature.

## 4. Conclusions

The anatase to rutile transition temperature decreased from 700 to 550 °C when the applied pressure varies from 76 to 400 MPa. However, the densification process occurred at a lower temperature under high pressure. Therefore, using high pressure SPS, we were able to densify the anatase powder: (i) At a sintering temperature as low as 450 °C with high applied pressure (400 MPa),

the anatase phase is retained with a very fine grain size (≈20 nm). A relative density up to 90% was achieved. (ii) At a higher sintering temperature, a phase transition from anatase to rutile concomitant with a fast grain growth was observed. A twinning of the rutile grains was induced by the phase transition and enhanced by pressure.

In general, to limit grain growth, the sintering temperature must be lowered when the high pressure process is used. Low-temperature sintering under high pressure is particularly interesting to sinter temperature-sensitive materials or metastable phases.

**Author Contributions:** Conceptualization, G.F. and S.L.F.; Formal analysis, C.S., T.G. and S.L.F.; Funding acquisition, S.D. and S.P.; Investigation, A.V., S.C., S.M. and N.B.; Project administration, S.P. and S.D.; Supervision, G.F.; Validation, S.P. and S.D.; Writing—original draft, S.C. and S.L.F.; Writing—review and editing, G.F. and S.L.F. All authors have read and agreed to the published version of the manuscript.

**Funding:** This work was supported by the LABEX iMUST (ANR-10-LABX-0064) of Université de Lyon, within the program "Investissements d'Avenir" (ANR-11-IDEX-0007) operated by the French National Research Agency (ANR).

**Acknowledgments:** The authors would particularly like to thank three people from the MATEIS laboratory: Bérangère Lesaint for her help in the preparation of thin foils for the TEM analysis, Laurent Gremillard and Sandrine Cardinal for their support in Rietveld analysis.

**Conflicts of Interest**: The authors declare no conflict of interest.

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
