# Peer review of "Effect of High Pressure Spark Plasma Sintering on the Densification of a Nb-Doped TiO2 Nanopowder"

_ceramics, doi:10.3390/ceramics3040041_

Round 1

Reviewer 1 Report

Dear Mr. Sorin Hadrian Petrescu

Assistant Editor

Ceramics

Manuscript ID: ceramics-993600

Title: Effect of high pressure – spark plasma sintering on the densification of a Nb-doped TiO2 nanopowder

Authors: Verchère Alexandre et al.

Reviewer Comments:

This paper addresses the effects of the SPS processing conditions (temp and pressure) on the microstructure and transformation of the Nb-doped TiO2 ceramics. The results obtained by the systematic work may be interested in the research field of SPS and give a new insight to the readers of the international journal of Ceramics. The content mentioned in this work is basically acceptable for the publication.

However, since a few statements are not clear and would confuse the readers, those should be carefully revised together with someone, who is familiar with English to help readers understand. For example, the authors should explain each figure carefully before the discussion to help readers understand. The authors just put the figures without enough explanation; the author should explain the figures what they want to claim using the figures. Some other comments are noted in the separated paper and in the pdf files.

As a result of the careful review, reviewer has to recommend to the editor that this paper can be accepted as a paper after the revision.

Minor comments

  1. Figure 2

What are the unidentified peaks in XRD?

  1. Page 5: line 136-138

The authors claimed that for 400 MPa, “a sharp jump at 546°C may correspond to the transition from the anatase to the rutile phases”.

On the other hand, 365°C is the sintering temperature of the anatase phase and 620°C is the sintering temperature of the rutile phase. This means that the anatase phase becomes dense below the transition temperature of 546°C and the rutile would be dense already. Why the sintering temperature of the rutile phase appears at 620°C once again?

  1. Pages, 11-12

Density, that is porosity, would influence the rate of grain growth due to the pinning effect. It would be better to refer the pore pinning effect.

  1. Pages, 11-12

First, reviewer feels that is would be better to refer to dynamic grain growth. If not, what is the difference between dynamic grain growth and the proposed mechanism?

Otherwise, before discuss the mechanism of the grain growth, the authors should discuss the rate of essential grain growth of the anatase and rutile phases.

Author Response

 We thank the Reviewer 1 for the carefully reading of our manuscript and her/his positive appreciation for its publication in Ceramics. We have considered and addressed her/his questions and suggestions which have enabled us to improve the presentation of our results and the discussion in the manuscript.  Our responses are written in the coverletter.

Reviewer 2 Report

Nice publication. Some quastions:

  1. Do you have any DLS maesurement of particle size distribution? If yes please add to the paper.
  2. The densities, you achieved by SPS, are not very high. Please add the explanation. Do you have any impurities coming from carbon on/in the sinter surface?
  3. Do you have calibration made on the piston displacement. If there is graphite foil between piston and powder what is the influence of graphite foil on displacement.
  4. SPS generates lots of stress to the material - do you have any larger cracks. Generally, From my experience materials are very brittle after SPS process.
  5. Please add first derivative to the dilatogram. It helps sometimes with discussion.
  6. Is SPS good for oxides' sintering?

Author Response

We thank the reviewer 2 for the reading of our manuscript and her/his positive appreciation of our work for publication in Ceramics. We have answered to her/his comments in the attached file.
